# Acylpyruvates and Their Heterocyclic Derivatives as Growth Regulators in *Chlorella vulgaris*

**DOI:** 10.3390/biotech14040090

**Published:** 2025-11-10

**Authors:** Anastasia D. Novokshonova, Pavel V. Khramtsov, Maksim V. Dmitriev, Ekaterina E. Khramtsova

**Affiliations:** 1Department of Chemistry, Perm State University, Ulitsa Bukireva, 15, Perm 614990, Russia; 2Institute of Ecology and Genetics of Microorganisms, Perm Federal Research Center, Ural Branch of Russian Academy of Sciences, Ulitsa Goleva, 13, Perm 614081, Russia; 3Department of Biology, Perm State University, Ulitsa Bukireva, 15, Perm 614990, Russia

**Keywords:** acylpyruvate, 1,4-benzoxazine, biomass, carotenoid, *Chlorella*, chlorophyll, heterocycle, lipid, microalgae

## Abstract

Acylpyruvate derivatives represent a promising yet underexplored class of compounds for modulating microalgal growth and metabolism. Inspired by the metabolic role of pyruvate and the diverse bioactivity of its acylated analogs, this study investigates the structure–activity relationship of a diverse library of 55 acylpyruvate-derived compounds for stimulation of the green microalga *Chlorella vulgaris*. The library, encompassing 12 chemotypes including acylpyruvic acids, their esters, and various heterocyclic derivatives, was screened for effects on *C. vulgaris* growth. Six compounds were identified as active ones that enhanced biomass production in a preliminary microassay. Notably, four of these active compounds were direct acylpyruvate derivatives, highlighting this scaffold as the most promising one. Conversely, a specific subclass, 1,4-benzoxazin-2-ones, exhibited potent, dose-dependent algicidal activity. Detailed assessment of the active compounds under scaled-up culture conditions revealed that while their effect on overall cell density was limited, several compounds significantly enhanced the intracellular content of valuable metabolites: one increased chlorophyll content by 17%, another elevated carotenoids by 40%, and a third boosted neutral lipid accumulation by 44%. Furthermore, control experiments confirmed that the bioactivity of *p*-ethoxybenzoylpyruvates, which showed the best biological activity, is inherent in the intact framework and is not mediated by their hydrolysis products. Our findings underscore the potential of acylpyruvates as versatile tools for the enhancement of metabolite production in microalgae and as potent candidates for the development of algicides.

## 1. Introduction

Pyruvate (Figure 1), a metabolite in glycolysis and the tricarboxylic acid cycle, serves as a critical node in cellular energy metabolism and biosynthetic pathways. Its enzymatic conversion to oxaloacetate—an important intermediate in tricarboxylic acid cycle [1], gluconeogenesis [2], and amino acid synthesis [3]—highlights the metabolic versatility of pyruvate-derived compounds. Acylation of pyruvate to afford acylpyruvates (Figure 1) introduces unique chemical properties that enhance its reactivity [4] and biological functionality, enabling interactions with key enzymes and signaling molecules [5]. Acylpyruvate derivatives may mimic or modulate the behavior of endogenous tricarboxylic acid cycle intermediates like oxaloacetate.

It is important to note that acylpyruvic acids and their esters bear a 1,3-dicarbonyl moiety that undergoes keto-enol tautomerization, with the enol form being predominant both in solution and in the solid state (Figure 1) [4,6].

Thus, acylpyruvate derivatives (Figure 1), a class of structurally modified pyruvate compounds, have gained increasing attention due to their diverse biological activities. They were studied in the development of antiviral (e.g., against HIV-1 [7], HCV [8], influenza [9], and SARS-CoV-2 [10]), antibacterial (e.g., targeting *Staphylococcus aureus* [11,12], *Mycobacterium tuberculosis* [13], *Streptococcus pneumoniae* [14]), antifungal (e.g., against *Candida albicans* [15], *Microsporum canis* [15], *Fusarium oxysporum* [16], *Helminthosporium turcicum* [16]), anti-inflammatory (e.g., as dual inhibitors of synthesis of both TNF-α and IL-6 in BEAS-2B cells [17]), neuroprotective (e.g., via inhibition of kynurenine metabolism [18]), and antitumor drugs (e.g., exhibiting cytotoxicity against MDA-MB-231 [19]), as well as in agriculture (e.g., to increase productivity and stress resistance in higher plants [20]) and metabolic research (e.g., for studying the urine [21] and fecal [22] metabolome). Their ability to inhibit key enzymes makes them promising candidates for pharmacology and biotechnology.

This study is a part of our broader project (for recent publications from this project, see [23,24]) aimed at identifying stimulators of microalgae growth. Consistent with this approach, which involves the chemical screening of novel compounds, several studies have already identified promising candidates capable of enhancing both biomass production and the accumulation of valuable metabolites, such as neutral lipids [25,26,27,28]. The green microalga *Chlorella vulgaris* was selected as the target species due to its significant biotechnological relevance, where it is widely utilized for applications such as water remediation, as well as in animal feed and human nutrition [29,30]. These applications require cost-effective and efficient methods to boost biomass production and promote the accumulation of specific metabolites, such as proteins for food-related purposes.

The search for effective chemical stimulants to enhance microalgal productivity has led to the investigation of diverse compound classes. Among the most studied are phytohormones, such as auxins and cytokinins (Figure 2), which have been shown to influence cell division and biomass yield in *Chlorella* cultures [25,27]. Similarly, simple organic acids like benzoic and salicylic acid (Figure 2) have been identified as signaling molecules that can improve cell growth [31].

However, a systematic investigation into the structure–activity relationship of a distinct class of compounds—acylpyruvates and their heterocyclic derivatives—has not been conducted for microalgae. Acylpyruvates, which integrate a central metabolic precursor (pyruvate) with a diverse array of acyl groups, represent a novel and structurally tunable scaffold. Unlike the aforementioned modulators, the potential of this specific chemical class to act as both growth stimulants and metabolic modulators in *C. vulgaris* remains largely unexplored and constitutes a significant gap in the literature.

Noteworthy, acylpyruvates and their derivatives remain largely unexplored in the context of microalgal biotechnology. Previous studies on algal growth stimulants have primarily focused on plant hormones, simple organic acids, or known metabolic intermediates [25,26,27,28,31,32]. However, a systematic investigation into the structure–activity relationship (SAR) of acylpyruvates and their derivatives has not been conducted for microalgae.

Therefore, this study is the first to evaluate the effects of different acylpyruvate derivatives on the growth and biochemical composition (pigments, proteins, carbohydrates, and neutral lipids) of *C. vulgaris* and perform an initial SAR analysis of a structurally diverse library of acylpyruvate derivatives and their heterocyclic analogs, targeting the stimulation of *C. vulgaris* growth and metabolite production. This approach moves beyond simply increasing biomass yield and explores the potential of these compounds to tailor the biochemical composition of microalgal cells for specific biotechnological applications.

## 2. Materials and Methods

### 2.1. Characterization of Compounds ***1b**, **2a**-**f**,**j**-**n**, **3a**-**i**, **4**, **6a**-**g**, **7a**-**d**, **8**, **9a**, **10a**-**q**, **11**, **13a**, **14**, **15**, **16***, and ***17*** Studied in Biological Assays

#### 2.1.1. General Information

^1^H and ^13^C NMR spectra (Appendix A) were acquired on a Bruker Avance III 400 HD spectrometer (Bruker BioSpin AG, Faellanden, Switzerland) (at 400 and 100 MHz, respectively) in CDCl_3_ or DMSO-*d*_6_, using TMS (in ^1^H NMR, 0.00 ppm) or solvent residual signals (in ^13^C NMR, 77.00 ppm for CDCl_3_, 39.52 ppm for DMSO-*d*_6_; in ^1^H NMR, 7.26 ppm for CDCl_3_, 2.50 ppm for DMSO-*d*_6_) as internal standards. IR spectra were recorded on a Perkin–Elmer Spectrum Two spectrometer (PerkinElmer Inc., Waltham, MA, USA) from mulls in mineral oil. Melting points were measured on a Mettler Toledo MP70 apparatus (Mettler-Toledo (MTADA), Schwerzenbach, Switzerland) or Khimlabpribor PTP apparatus (Khimlabpribor, Klin, USSR). Elemental analyses were carried out on a Vario MICRO Cube analyzer (Elementar Analysensysteme GmbH, Langenselbold, Germany). The single-crystal X-ray analyses of compounds **2n**, **6f**, and **14** were performed on an Xcalibur Ruby diffractometer (Agilent Technologies, Wroclaw, Poland). The empirical absorption correction was introduced by using the multi-scan method with the SCALE3 ABSPACK algorithm [33]. Using OLEX2 [34], the structures were solved with the SHELXS [35] or SHELXT [36] program and refined by the full-matrix least-squares minimization in the anisotropic approximation for all non-hydrogen atoms with the SHELXL [37] program. Hydrogen atoms were positioned geometrically and refined using a riding model. Hydrogen atoms were positioned geometrically and refined using a riding model.

Benzene for operation *ix*, toluene for operations *v*, *vii*, *x*, Et_2_O for operation *i*, and 1,4-dioxane for operation *viii* were distilled over Na before use. Procedures *i*, *v*, *vii-x* were performed in an oven-dried glassware. All other solvents and reagents were purchased from commercial vendors and were used as received.

*o*-Aminophenol **9a** was purchased from Acros Organics (99% purity) and used in biological assays without additional purification. 4-Ethoxybenzoic acid **17** was purchased from Sigma-Aldrich (99% purity) and used in biological assays without additional purification. 4′-Ethoxyacetophenone **1b** was purchased from Sigma-Aldrich (98% purity) and used in biological assays without additional purification.

#### 2.1.2. Synthesis and Analytic Data of Compounds **2a**-**f**,**j**-**n**, **3a**-**i**, **4**, **6a**-**g**, **7a**-**d**, **8**, **10a**-**q**, **11**, **13a**, **14**, **15**, and **16**


**General procedure for compounds 2**
**a-f, j-m.**


A homogenous solution of a corresponding methylketone **1a**-**f**,**j**-**m** (0.5 mol) and diethyl oxalate (0.5 mol, 68.0 mL) (for solid methylketones **1g**,**k**-**m**, addition of methanol or 1,4-dioxane (about 100 mL) can be required to obtain a homogenous solution of them in diethyl oxalate) was added to a stirred cold (~5 °C) freshly prepared solution of sodium methoxide (0.5 mol, 27.0 g) in methanol (100 mL). Then, the reaction mixture was allowed to reach room temperature (the reaction mixture solidified). In 24 h, the reaction mixture was treated with HCl (50 mL of conc. HCl in 500 mL of water), and the resulting suspension was stirred overnight at room temperature. The formed precipitate was filtered off, washed with water (500 mL), dried and recrystallized from an appropriate solvent to result in a corresponding methyl acylpyruvate **2a**-**f**,**j**-**m**.

*Methyl (Z)-2-hydroxy-4-oxo-4-phenylbut-2-enoate (**2a**)* [38]. Yield: 80 g (78%); beige-white solid; mp 58–60 °C (petroleum ether {bp 70–100 °C}/DCM, 1:5 *v*/*v*) (lit. 57–58 °C [38]). ^1^H NMR (400 MHz, CDCl_3_) δ: 15.23 (br. s, 1 H), 7.99 (m, 2 H), 7.60 (m, 1 H), 7.50 (m, 2 H), 7.07 (s, 1 H), 3.94 (s, 3 H) ppm.

*Methyl (Z)-4-(4-ethoxyphenyl)-2-hydroxy-4-oxobut-2-enoate (**2b**).* Yield: 90 g (72%); beige-white solid; mp 81–83 °C (isooctane). ^1^H NMR (400 MHz, CDCl_3_) δ: 15.41 (br. s, 1 H), 7.97 (m, 2 H), 7.01 (s, 1 H), 6.95 (m, 2 H), 4.12 (q, *J* = 6.8 Hz, 2 H), 3.93 (s, 3 H), 1.45 (t, *J* = 6.8 Hz, 3 H) ppm. ^13^C NMR (400 MHz, CDCl_3_): δ = 190.3, 167.8, 163.9, 163.0, 130.3, 127.5, 114.7, 97.8, 64.0, 53.0, 14.6 ppm. IR (mineral oil): 3200, 1729, 1603 cm^−1^. Anal. Calcd (%) for C_13_H_14_O_5_: C 62.39; H 5.64. Found: C 62.69; H 5.38.

*Methyl (Z)-2-hydroxy-4-(4-methoxyphenyl)-4-oxobut-2-enoate (**2c**)* [38]. Yield: 84 g (71%); beige-white solid; mp 85–88 °C (chloroform) (lit. 92–93 °C [38]). ^1^H NMR (400 MHz, CDCl_3_) δ: 15.37 (br. s, 1 H), 7.98 (m, 2 H), 7.01 (s, 1 H), 6.97 (m, 2 H), 3.93 (s, 3 H), 3.88 (s, 3 H) ppm.

*Methyl (Z)-4-(furan-2-yl)-2-hydroxy-4-oxobut-2-enoate (**2d**)* [39]. Yield: 50 g (51%); beige-white solid; mp 82–84 °C (MeOH) (lit. 95–96 °C [39]). ^1^H NMR (400 MHz, CDCl_3_) δ: ^1^H NMR (400 MHz, CDCl_3_) δ: 14.43 (br. s, 1 H), 7.67 (m, 1 H), 7.34 (m, 1 H), 6.94 (s, 1 H), 6.61 (m, 1 H), 3.93 (s, 3 H) ppm.

*Methyl (Z)-2-hydroxy-4-oxo-4-(thiophen-2-yl)but-2-enoate (**2e**)* [40]. Yield: 73 g (69%); beige-white solid; mp 83–85 °C (MeOH) (lit. 79 °C [40]). ^1^H NMR (400 MHz, CDCl_3_) δ: 14.58 (br. s, 1 H), 7.85 (m, 1 H), 7.74 (m, 1 H), 7.19 (m, 1 H), 6.92 (s, 1 H), 3.94 (s, 3 H) ppm.

*Methyl (Z)-4-(4-chlorophenyl)-2-hydroxy-4-oxobut-2-enoate (**2f**)* [41]. Yield: 99 g (85%); beige-white solid; mp 95–97 °C (petroleum ether {bp 70–100 °C}/DCM, 1:5 *v*/*v*) (lit. 110–112 °C [41]). ^1^H NMR (400 MHz, CDCl_3_) δ: 15.12 (br. s, 1 H), 7.93 (m, 2 H), 7.48 (m, 2 H), 7.02 (s, 1 H), 3.94 (s, 3 H) ppm.

*Methyl (Z)-4-(4-fluorophenyl)-2-hydroxy-4-oxobut-2-enoate (**2j**)* [42]. Yield: 99 g (88%); beige-white solid; mp 109–110 °C (petroleum ether {bp 70–100 °C}) (lit. 109–110 °C [42]). ^1^H NMR (400 MHz, CDCl_3_) δ: 15.15 (br. s, 1 H), 8.03 (m, 2 H), 7.18 (m, 2 H), 7.03 (s, 1 H), 3.95 (s, 3 H) ppm.

*Methyl (Z)-2-hydroxy-4-(4-nitrophenyl)-4-oxobut-2-enoate (**2k**)* [38]. Yield: 77 g (61%); beige-white solid; mp 155–156 °C (*n*-butyl acetate) (lit. 161–162 °C [38]). ^1^H NMR (400 MHz, CDCl_3_) δ: 14.89 (br. s, 1 H), 8.33 (m, 2 H), 8.13 (m, 2 H), 7.08 (s, 1 H), 3.96 (s, 3 H) ppm.

*Methyl (Z)-2-hydroxy-4-(3-nitrophenyl)-4-oxobut-2-enoate (**2l**)* [38]. Yield: 72 g (65%); beige-white solid; mp 112–114 °C (MeOH) (lit. 112–113 °C [38]). ^1^H NMR (400 MHz, CDCl_3_) δ: 14.94 (br. s, 1 H), 8.81 (m, 1 H), 8.45 (m, 1 H), 8.31 (m, 1 H), 7.73 (m, 1 H), 7.11 (s, 1 H), 3.97 (s, 3 H) ppm.

*Methyl (Z)-4-(4-cyanophenyl)-2-hydroxy-4-oxobut-2-enoate (**2m**)* [43]. Yield: 65 g (56%); beige-white solid; mp 170–175 °C (EtOH) (lit. 170–175 °C [43]). ^1^H NMR (400 MHz, CDCl_3_) δ: 14.93 (br. s, 1 H), 8.08 (m, 2 H), 7.80 (m, 2 H), 7.06 (s, 1 H), 3.96 (s, 3 H) ppm.

*Diethyl (2Z,5Z)-2,6-dihydroxy-4-oxohepta-2,5-dienedioate (**2n**)* [44]. A solution of acetone **1n** (0.1 mol, 7.3 mL) and diethyl oxalate (0.11 mol, 15.0 mL) was added to a stirred warm (~60 °C) freshly prepared solution of sodium ethoxide (0.1 mol, 6.8 g) in absolute ethanol (60 mL). In 3 min, hot (~80 °C) freshly prepared solution of sodium ethoxide (0.1 mol, 6.8 g) in absolute ethanol (60 mL) and diethyl oxalate (0.11 mol, 15.0 mL) were added simultaneously to the reaction mixture. The reaction mixture was stirred and heated (~80 °C) for 15 min (the reaction mixture solidified). Next day, the reaction mixture was heated in an oil bath (110°) until ethanol had distilled. Then, the reaction mixture was allowed to reach room temperature. After that, the solid was treated with HCl (a mixture of 65 mL of conc. HCl and 150 g of cracked ice). Yellow suspension of compound **2n** was filtered off and washed with cold water, dried and recrystallized from ethanol. Yield: 12 g (46%); yellow solid; mp 98–101 °C (EtOH) (lit. 98–100 °C [44]). ^1^H NMR (400 MHz, CDCl_3_) δ: 13.16 (br. s, 2 H), 6.34 (s, 2 H), 4.34 (q, *J* = 6.8 Hz, 4 H), 1.36 (t, *J* = 7.1 Hz, 6 H) ppm. Crystal structure of compound **2n** was deposited at the Cambridge Crystallographic Data Centre with the deposition number CCDC 2486572.


**General procedure for compounds 3**
**a-h.**


A corresponding acylpyruvate **2a**-**h** (0.05 mol) was stirred in a solution of NaOH (0.17 mol, 6.8 g) in water (200 mL) for 1 h. Then, the reaction mixture was treated with HCl (17 mL of conc. HCl in 100 mL of water), and the resulting suspension was stirred overnight at room temperature. The formed precipitate was filtered off, washed with water (100 mL), dried and recrystallized from an appropriate solvent to result in a corresponding acylpyruvic acid **3a**-**h.**

*(Z)-2-Hydroxy-4-oxo-4-phenylbut-2-enoic acid (**3a**)* [38]. Yield: 9 g (92%); beige-white solid; mp 149–150 °C (*n*-butyl acetate) (lit. 154–157 °C [38]). ^1^H NMR (400 MHz, CDCl_3_) δ: 15.06 (br. s, 1 H), 8.01 (m, 2 H), 7.64 (m, 1 H), 7.53 (m, 2 H), 7.18 (s, 1 H) ppm.

*(Z)-4-(4-Ethoxyphenyl)-2-hydroxy-4-oxobut-2-enoic acid (**3b**).* Yield: 10 g (85%); yellowish-white solid; mp 81–83 °C (*n*-butyl acetate). ^1^H NMR (400 MHz, CDCl_3_) δ: 15.06 (br. s, 1 H), 7.99 (m, 2 H), 7.09 (s, 1 H), 6.98 (m, 2 H), 4.14 (q, *J* = 7.2 Hz, 2 H), 1.46 (t, *J* = 7.1 Hz, 3 H) ppm. ^13^C NMR (400 MHz, CDCl_3_): δ = 187.0, 172.9, 164.3, 162.3, 130.4, 125.6, 114.9, 94.8, 64.1, 14.6 ppm. IR (mineral oil): 3500, 1691, 1605 cm^−1^. Anal. Calcd (%) for C_12_H_12_O_5_: C 61.02; H 5.12. Found: C 60.78; H 5.23.

*(Z)-2-Hydroxy-4-(4-methoxyphenyl)-4-oxobut-2-enoic acid (**3c**)* [38]. Yield: 9 g (82%); yellowish-white solid; mp 161–163 °C (*n*-butyl acetate) (lit. 157–158 °C [38]). ^1^H NMR (400 MHz, CDCl_3_) δ: 8.00 (m, 2 H), 7.09 (s, 1 H), 7.00 (m, 2 H), 3.91 (s, 3 H) ppm.

*(Z)-4-(Furan-2-yl)-2-hydroxy-4-oxobut-2-enoic acid (**3d**)* [45]. Yield: 5 g (58%); yellowish-white solid; mp 154–156 °C (*n*-butyl acetate) (lit. 145–148 °C [45]). ^1^H NMR (400 MHz, CDCl_3_) δ: 7.71 (m, 1 H), 7.38 (m, 1 H), 7.04 (s, 1 H), 6.65 (m, 1 H) ppm.

*(Z)-2-Hydroxy-4-oxo-4-(thiophen-2-yl)but-2-enoic acid (**3e**)* [45]. Yield: 7 g (73%); yellowish-white solid; mp 175–177 °C (*n*-butyl acetate) (lit. 170–173 °C [45]). ^1^H NMR (400 MHz, CDCl_3_) δ: 7.89 (m, 1 H), 7.77 (m, 1 H), 7.21 (m, 1 H), 7.00 (s, 1 H) ppm.

*(Z)-4-(4-Chlorophenyl)-2-hydroxy-4-oxobut-2-enoic acid (**3f**)* [46]. Yield: 10 g (91%); yellowish-white solid; mp 166–167 °C (*n*-butyl acetate) (lit. 164–165 °C [46]). ^1^H NMR (400 MHz, CDCl_3_) δ: 15.04 (br. s, 1 H), 7.95 (m, 2 H), 7.50 (m, 2 H), 7.13 (s, 1 H) ppm.

*(Z)-4-(4-Bromophenyl)-2-hydroxy-4-oxobut-2-enoic acid (**3g**)* [46]. Yield: 13 g (96%); white solid; mp 174–176 °C (*n*-butyl acetate) (lit. 163–164 °C [46]). ^1^H NMR (400 MHz, CDCl_3_) δ: 14.90 (br. s, 1 H), 7.87 (m, 2 H), 7.67 (m, 2 H), 7.13 (s, 1 H) ppm.

*(Z)-2-Hydroxy-4-(naphthalen-2-yl)-4-oxobut-2-enoic acid (**3h**)* [47]. Yield: 9 g (73%); yellowish-white solid; mp 167–169 °C (*n*-butyl acetate) (lit. 170–171 °C [47]). ^1^H NMR (400 MHz, CDCl_3_) δ: 15.26 (br. s, 1 H), 8.60 (m, 1 H), 8.02–7.90 (m, 4 H), 7.67–7.57 (m, 2 H), 7.33 (s, 1 H) ppm.

*(Z)-2-Hydroxy-5,5-dimethyl-4-oxohex-2-enoic acid (**3i**)* [48]. Methyl pivaloylpyruvate **2i** (0.05 mol, 10 g) was stirred in a solution of NaOH (0.17 mol, 6.8 g) in water (200 mL) for 1 h. Then, the reaction mixture was treated with HCl (17 mL of conc. HCl in 100 mL of water), and the resulting suspension was stirred for 2 h at room temperature. Compound **3i** was extracted from the resulting mixture with DCM (50 × 25 × 25 mL)**.** Then, the extract was evaporated to dryness, and the resulting solid of compound **3i** was recrystallized from chloroform. Yield: 7 g (79%); white solid; mp 51–53 °C (chloroform) (lit. 54–55 °C [48]). ^1^H NMR (400 MHz, CDCl_3_) δ: 14.70 (br. s, 1 H), 6.62 (s, 1 H), 1.24 (s, 9 H) ppm.

*4-Oxo-4H-pyran-2,6-dicarboxylic acid (chelidonic acid) (**4**)* [44]. Diethyl (2*Z*,5*Z*)-2,6-dihydroxy-4-oxohepta-2,5-dienedioate **2n** (3.9 μmol, 1 g) was heated with conc. HCl (15 mL) on the steam bath for 24 h. After cooling to room temperature, the precipitate of compound **4** was filtered off, washed with water (20 mL), and dried. Yield: 0.43 g (61%); beige solid; mp 256–257 °C (water) (lit. 257 °C [44]). ^1^H NMR (400 MHz, DMSO-*d*_6_) δ: 13.46 (br. s, 2 H), 6.95 (s, 2 H) ppm.


**General procedure for compounds 6**
**a-g**
**.**


A mixture of a corresponding acylpyruvate **2a**,**c**,**p**,**r** (0.01 mol) and a corresponding *o*-phenylenediamine **5a**-**c** (0.01 mol) was refluxed in ethanol (10–20 mL) for 2 h. Then, the reaction mixture was cooled to room temperature. The formed precipitate was filtered off, washed with ethanol (10 mL) and recrystallized from an appropriate solvent to result in the corresponding compound **6a**-**g.**

*Ethyl (Z)-2-(3-oxo-3,4-dihydroquinoxalin-2(1H)-ylidene)acetate (**6a**)* [49]. Yield: 1.9 g (82%); orange solid; mp 208–210 °C (1,4-dioxane) (lit. 218 °C [49]). ^1^H NMR (400 MHz, DMSO-*d*_6_) δ: 11.65 (s, 1 H), 11.04 (s, 1 H), 7.37 (m, 1 H), 7.08–6.99 (m, 3 H), 5.51 (s, 1 H), 4.16 (q, *J* = 7.0 Hz, 2 H), 1.25 (t, *J* = 7.0 Hz, 3 H) ppm.

*(Z)-3-(2-Oxo-2-phenylethylidene)-3,4-dihydroquinoxalin-2(1H)-one (**6b**)* [50]. Yield: 2.2 g (85%); orange solid; mp 265–268 °C (1,4-dioxane) (lit. 266–267 °C [50]). ^1^H NMR (400 MHz, DMSO-*d*_6_) δ: 13.67 (s, 1 H), 11.98 (s, 1 H), 7.99 (m, 2 H), 7.61–7.49 (m, 4 H), 7.15–7.12 (m, 3 H), 6.83 (s, 1 H) ppm.

*(Z)-3-((E)-2-Oxo-4-phenylbut-3-en-1-ylidene)-3,4-dihydroquinoxalin-2(1H)-one (**6c**)* [51]. Yield: 2.1 g (71%); orange solid; mp 245–247 °C (1,4-dioxane) (lit. 247 °C [51]). ^1^H NMR (400 MHz, DMSO-*d*_6_) δ: 13.84 (s, 1 H), 11.95 (s, 1 H), 7.71 (m, 2 H), 7.53 (d, *J* = 15.8 Hz, 1 H), 7.45–7.37 (m, 4 H), 7.17–7.13 (m, 3 H), 7.10 (d, *J* = 15.8 Hz, 1 H), 6.38 (s, 1 H) ppm.

*(Z)-3-(2-Oxo-2-phenylethylidene)-1-phenyl-3,4-dihydroquinoxalin-2(1H)-one (**6d**)* [52]. Yield: 2.8 g (81%); orange solid; mp 217–219 °C (toluene) (lit. 217–218 °C [52]). ^1^H NMR (400 MHz, DMSO-*d*_6_) δ: 13.86 (s, 1 H), 8.00 (m, 2 H), 7.68–7.52 (m, 8 H), 7.45 (m, 2 H), 7.20 (m, 1 H), 7.05 (m, 1 H), 6.88 (s, 1 H), 6.37 (m, 1 H) ppm.

*(Z)-3-(2-(4-Methoxyphenyl)-2-oxoethylidene)-1-phenyl-3,4-dihydroquinoxalin-2(1H)-one (**6e**)* [52]. Yield: 3.2 g (86%); orange solid; mp 217–219 °C (toluene) (lit. 218–219 °C [52]). ^1^H NMR (400 MHz, DMSO-*d*_6_) δ: 13.77 (s, 1 H), 7.98 (m, 2 H), 7.65 (m, 2 H), 7.58 (m, 2 H), 7.44 (m, 2 H), 7.18 (m, 1 H), 7.07 (m, 2 H), 7.01 (m, 1 H), 6.84 (s, 1 H), 6.35 (m, 1 H), 3.85 (s, 3 H) ppm.

*(Z)-3-((E)-2-Oxo-4-phenylbut-3-en-1-ylidene)-1-phenyl-3,4-dihydroquinoxalin-2(1H)-one (**6f**)* [53]. Yield: 2.9 g (80%); orange solid; mp 204–205 °C (toluene) (lit. 204–205 °C [53]). ^1^H NMR (DMSO-*d*_6_, 400 MHz) δ: 14.05 (s, 1 H), 7.72 (m, 2 H), 7.65 (m, 2 H), 7.56 (m, 3 H), 7.43 (m, 5 H), 7.18 (m, 2 H), 7.07 (m, 1 H), 6.43 (s, 1 H), 6.37 (m, 1 H) ppm. ^13^C NMR (DMSO-*d*_6_, 100 MHz) δ: 184.5, 155.1, 146.1, 138.4, 136.3, 135.0, 130.0, 130.0, 129.6, 129.0, 128.8, 128.6, 128.0, 127.6, 125.4, 124.2, 123.9, 118.0, 115.4, 95.1 ppm. IR (mineral oil): 3107, 1674 cm^−1^. Anal. Calcd (%) for C_24_H_18_N_2_O_2_: C 78.67; H 4.95; N 7.65. Found: C 78.93; H 5.11; N 7.68. Crystal structure of compound **6f** was deposited at the Cambridge Crystallographic Data Centre with the deposition number CCDC 2010681 (Refcode ZURBOG) [53].

*(Z)-3-(2-(4-Methoxyphenyl)-2-oxoethylidene)-1-methyl-3,4-dihydroquinoxalin-2(1H)-one (**6g**)* [54]. Yield: 2.86 g (93%); orange solid; mp 173–173 °C (toluene) (lit. 173–173 °C [54]). ^1^H NMR (400 MHz, DMSO-*d*_6_) δ: 13.75 (s, 1 H), 7.96 (m, 2 H), 7.50 (m, 1 H), 7.40 (m, 1 H), 7.22 (m, 2 H), 7.08 (m, 2 H), 6.82 (s, 1 H), 3.85 (s, 3 H), 3.60 (s, 3 H) ppm.


**General procedure for compounds 7**
**a-d.**


To a cold (~5 °C) stirring suspension of a corresponding acylpyruvic acid **3h**,**j**,**k**,**l** (5 mmol) and *o*-aminothiophenol (5.1 mmol, 0.5 mL) in toluene (8 mL), a solution of DCC (5 mmol, 1 g) in toluene (3 mL) was added. The mixture was allowed to reach room temperature overnight. Then, the formed yellow precipitate was filtered off under vacuum and washed with toluene (50–100 mL) to remove precipitated orange crystals of the target compound. The collected orange mother liquor from filtration was evaporated to dryness on a rotavap. The solid residue was stirred with ethanol (10 mL) for 1 h to remove dicylohexylurea. Then, the orange precipitate was filtered off and recrystallized from toluene (5–10 mL) to give the corresponding compound **7a**-**d**.

*(Z)-3-(2-Oxo-2-(p-tolyl)ethylidene)-3,4-dihydro-2H-benzo[b][1,4]thiazin-2-one (**7a**)* [55]. Yield: 0.67 g (45%); orange solid; mp 176–178 °C (toluene) (lit. 176–178 °C [55]). ^1^H NMR (400 MHz, DMSO-*d*_6_) δ: 13.81 (s, 1 H), 7.91 (m, 2 H), 7.48–7.32 (m, 5 H), 7.20 (m, 1 H), 6.72 (s, 1 H), 2.39 (s, 3 H) ppm.

*(Z)-3-(2-(3-Methoxyphenyl)-2-oxoethylidene)-3,4-dihydro-2H-benzo[b][1,4]thiazin-2-one (**7b**).* Yield: 0.44 g (28%); orange solid; mp 156–158 °C (toluene). ^1^H NMR (DMSO-*d*_6_, 400 MHz) δ: 13.84 (s, 1 H), 7.59 (m, 1 H), 7.49–7.39 (m, 5 H), 7.21 (m, 2 H), 6.71 (s, 1 H), 3.84 (s, 3 H) ppm. ^13^C NMR (DMSO-*d*_6_, 100 MHz) δ: 190.0, 181.5, 159.5, 142.8, 139.7, 129.9, 129.6, 128.5, 126.2, 123.7, 119.6, 118.7, 118.4, 116.7, 111.7, 88.3, 55.2 ppm. IR (mineral oil): 3134, 1655 cm^−1^. Anal. Calcd (%) for C_17_H_13_NO_3_S: C 65.58; H 4.21; N 4.50. Found: C 65.79; H 4.11; N 4.68.

*(Z)-3-(2-(4-Fluorophenyl)-2-oxoethylidene)-3,4-dihydro-2H-benzo[b][1,4]thiazin-2-one (**7c**)* [55]. Yield: 0.63 g (42%); orange; mp 175–177 °C (toluene) (lit. 175–177 °C [55]). ^1^H NMR (400 MHz, DMSO-*d*_6_) δ: 13.79 (s, 1 H), 8.09 (m, 2 H), 7.50–7.47 (m, 2 H), 7.43–7.41 (m, 1 H), 7.35 (m, 2 H), 7.24–7.20 (m, 1 H), 6.72 (s, 1 H) ppm.

*(Z)-3-(2-(Naphthalen-2-yl)-2-oxoethylidene)-3,4-dihydro-2H-benzo[b][1,4]thiazin-2-one (**7d**)* [55]. Yield: 0.81 g (41%); orange; mp 201–203 °C (toluene) (lit. 201–203 °C [55]). ^1^H NMR (400 MHz, DMSO-*d*_6_) δ: 13.93 (s, 1 H), 8.70 (m, 1 H), 8.19 (m, 1 H), 8.07 (m, 2 H), 8.00 (m, 1 H), 7.68–7.60 (m, 2 H), 7.50 (m, 2 H), 7.43 (m, 1 H), 7.23 (m, 1 H), 6.95 (s, 1 H) ppm.

*(Z)-3-(2-Oxo-2-(p-tolyl)ethylidene)piperazin-2-one (**8**)* [56]. A mixture of 4-toluoylpyruvic acid **3j** (5.0 mmol, 1.00 g) and 1,2-ethylenediamine (5.0 mmol, 333 μL) was refluxed in ethanol (10 mL) for 2 h. Then, the reaction mixture was cooled to room temperature. The formed precipitate was filtered off, washed with ethanol (5 mL) and recrystallized from ethanol (15 mL) to result in compound **8**. Yield: 0.85 g (74%); yellow solid; mp 223–224 °C (EtOH) (lit. 223–224 °C [56]). ^1^H NMR (400 MHz, CDCl_3_) δ: 10.63 (br. s, 1 H), 7.87 (m, 2 H), 7.23 (m, 2 H), 6.77 (s, 1 H), 6.65 (br. s, 1 H), 3.59 (m, 2 H), 3.54 (m, 2 H) ppm.


**General procedure for compounds 10**
**a-n.**


A mixture of the corresponding acylpyruvate **2a**,**i**,**o**,**p** (4.0 mmol) and the corresponding *o*-aminophenol **9a-h** (4.0 mmol) was refluxed in ethanol (10 mL) for 2 h. Then, the reaction mixture was cooled to room temperature. The formed precipitate was filtered off, washed with ethanol (5 mL) and recrystallized from an appropriate solvent to result in the corresponding compound **10a**-**n**.

*(Z)-3-(2-Oxo-2-phenylethylidene)-3,4-dihydro-2H-benzo[b][1,4]oxazin-2-one (**10a**)* [57]. Yield: 0.88 g (83%); yellow solid; mp 207–208 °C (toluene) (lit. 203–205 °C [57]). ^1^H NMR (400 MHz, CDCl_3_) δ: 13.06 (s, 1 H), 8.01 (m, 2 H), 7.55 (m, 1 H), 7.48 (m, 2 H), 7.19 (m, 2 H), 7.11 (m, 2 H), 7.06 (s, 1 H) ppm.

*(Z)-3-(3,3-Dimethyl-2-oxobutylidene)-3,4-dihydro-2H-benzo[b][1,4]oxazin-2-one (**10b**)* [42]. Yield: 0.81 g (83%); yellow solid; mp 86–88 °C (ethanol) (lit. 86–88 °C [42]). ^1^H NMR (400 MHz, DMSO-*d*_6_) δ: 12.33 (s, 1 H), 7.49 (m, 1 H), 7.19 (m, 2 H), 7.08 (m, 1 H), 6.36 (s, 1 H), 1.18 (s, 9 H) ppm.

*(Z)-2-(3,3-Dimethyl-2-oxobutylidene)-1,2-dihydro-3H-naphtho[2,1-b][1,4]oxazin-3-one (**10c**)* [42]. 1-Aminonaphthalen-2-ol **9b** hydrochloride was utilized in general procedure instead of free 1-aminonaphthalen-2-ol **9b**. Yield: 0.74 g (63%); yellow solid; mp 168–169 °C (ethanol) (lit. 168–169 °C [42]). ^1^H NMR (400 MHz, DMSO-*d*_6_) δ: 13.75 (s, 1 H), 7.99 (m, 1 H), 7.87 (m, 1 H), 7.74 (m, 1 H), 7.70 (m, 1 H), 7.59 (m, 1 H), 7.41 (m, 1 H), 6.44 (s, 1 H), 1.24 (s, 9 H) ppm.

*(Z)-3-(3,3-Dimethyl-2-oxobutylidene)-7-nitro-3,4-dihydro-2H-benzo[b][1,4]oxazin-2-one (**10d**)* [42]. Yield: 0.82 g (81%); yellow solid; mp 201–202 °C (ethanol) (lit. 201–202 °C [42]). ^1^H NMR (400 MHz, DMSO-*d*_6_) δ: 12.22 (s, 1 H), 8.03 (m, 2 H), 7.76 (m, 1 H), 6.50 (s, 1 H), 1.19 (s, 9 H) ppm.

*(Z)-3-(3,3-Dimethyl-2-oxobutylidene)-6-nitro-3,4-dihydro-2H-benzo[b][1,4]oxazin-2-one (**10e**)* [42]. Yield: 0.84 g (733%); yellow solid; mp 194–195 °C (ethanol) (lit. 194–195 °C [42]). ^1^H NMR (400 MHz, DMSO-*d*_6_) δ: 12.19 (s, 1 H), 8.63 (m, 1 H), 7.89 (m, 1 H), 7.39 (m, 1 H), 6.43 (s, 1 H), 1.19 (s, 9 H) ppm.

*(Z)-3-(3,3-Dimethyl-2-oxobutylidene)-6-methyl-3,4-dihydro-2H-benzo[b][1,4]oxazin-2-one (**10f**)* [42]. Yield: 0.84 g (81%); yellow solid; mp 142–143 °C (ethanol) (lit. 142–143 °C [42]). ^1^H NMR (400 MHz, DMSO-*d*_6_) δ: 12.31 (s, 1 H), 7.26 (m, 1 H), 7.08 (m, 1 H), 6.89 (m, 1 H), 6.34 (s, 1 H), 2.28 (s, 3 H), 1.17 (s, 9 H) ppm.

*(Z)-3-(3,3-Dimethyl-2-oxobutylidene)-5-methyl-3,4-dihydro-2H-benzo[b][1,4]oxazin-2-one (**10g**)* [42]. Yield: 0.86 g (83%); yellow solid; mp 121–122 °C (ethanol) (lit. 121–122 °C [42]). ^1^H NMR (400 MHz, DMSO-*d*_6_) δ: 12.77 (s, 1 H), 7.09 (m, 2 H), 7.02 (m, 1 H), 6.39 (s, 1 H), 2.34 (s, 3 H), 1.19 (s, 9 H) ppm.

*(Z)-6-Chloro-3-(4-methyl-2-oxopentylidene)-3,4-dihydro-2H-benzo[b][1,4]oxazin-2-one (**10h**)* [42]. Yield: 0.84 g (75%); yellow solid; mp 135–136 °C (ethanol) (lit. 135–136 °C [42]). ^1^H NMR (400 MHz, DMSO-*d*_6_) δ: 12.07 (s, 1 H), 7.71 (m, 1 H), 7.20 (m, 1 H), 7.07 (m, 1 H), 6.17 (s, 1 H), 2.41 (d, *J* 6.8 Hz, 2 H), 2.09 (n, *J* 6.8 Hz, 1 H), 0.93 (d, *J* 6.6 Hz, 6 H) ppm.

*(Z)-6-Methyl-3-(4-methyl-2-oxopentylidene)-3,4-dihydro-2H-benzo[b][1,4]oxazin-2-one (**10i**)* [42]. Yield: 0.81 g (72%); yellow solid; mp 89–90 °C (ethanol) (lit. 89–90 °C [42]). ^1^H NMR (400 MHz, DMSO-*d*_6_) δ: 12.25 (s, 1 H), 7.26 (m, 1 H), 7.09 (m, 1 H), 6.89 (m, 1 H), 6.13 (s, 1 H), 2.39 (d, *J* 7.1 Hz, 2 H), 2.28 (s, 3 H), 2.09 (n, *J* 6.7 Hz, 1 H), 0.92 (d, *J* 6.6 Hz, 6 H) ppm.

*(Z)-5-Methyl-3-(4-methyl-2-oxopentylidene)-3,4-dihydro-2H-benzo[b][1,4]oxazin-2-one (**10j**)* [42]. Yield: 0.85 g (76%); yellow solid; mp 99 °C (ethanol) (lit. 99 °C [42]). ^1^H NMR (400 MHz, DMSO-*d*_6_) δ: 12.76 (s, 1 H), 7.11 (m, 2 H), 7.03 (m, 1 H), 6.19 (s, 1 H), 2.42 (d, *J* 7.1 Hz, 2 H), 2.34 (s, 3 H), 2.11 (n, *J* 6.8 Hz, 1 H), 0.93 (d, *J* 6.8 Hz, 6 H) ppm.

*(Z)-6-Bromo-3-(4-methyl-2-oxopentylidene)-3,4-dihydro-2H-benzo[b][1,4]oxazin-2-one (**10k**)* [42]. Yield: 0.92 g (71%); yellow solid; mp 130–131 °C (ethanol) (lit. 130–131 °C [42]). ^1^H NMR (400 MHz, DMSO-*d*_6_) δ: 12.06 (s, 1 H), 7.84 (m, 1 H), 7.20 (m, 1 H), 7.14 (m, 1 H), 6.17 (s, 1 H), 2.41 (d, *J* 7.1 Hz, 2 H), 2.09 (n, *J* 6.8 Hz, 1 H), 0.93 (d, *J* 6.6 Hz, 6 H) ppm.

*(Z)-3-(4-Methyl-2-oxopentylidene)-3,4-dihydro-2H-benzo[b][1,4]oxazin-2-one (**10l**)* [42]. Yield: 0.77 g (79%); yellow solid; mp 108–110 °C (ethanol) (lit. 108–110 °C [42]). ^1^H NMR (400 MHz, DMSO-*d*_6_) δ: 12.25 (s, 1 H), 7.48 (m, 1 H), 7.19 (m, 2 H), 7.08 (m, 1 H), 6.14 (s, 1 H), 2.40 (d, *J* 7.1 Hz, 2 H), 2.10 (n, *J* 6.8 Hz, 1 H), 0.93 (d, *J* 6.8 Hz, 6 H) ppm.

*Ethyl (Z)-2-(2-oxo-2H-benzo[b][1,4]oxazin-3(4H)-ylidene)acetate (**10m**)* [42]. Yield: 0.63 g (71%); yellow solid; mp 110–112 °C (ethanol) (lit. 110–112 °C [42]). ^1^H NMR (400 MHz, CDCl_3_) δ: 10.69 (s, 1 H), 7.14 (m, 2 H), 7.00 (m, 2 H), 5.93 (s, 1 H), 4.25 (q, *J* 7.2 Hz, 2 H), 1.33 (t, *J* 7.2 Hz, 3 H) ppm.

*Ethyl (Z)-2-(6-bromo-2-oxo-2H-benzo[b][1,4]oxazin-3(4H)-ylidene)acetate (**10n**)* [42]. Yield: 0.97 g (78%); yellow solid; mp 137–139 °C (ethanol) (lit. 137–139 °C [42]). ^1^H NMR (400 MHz, DMSO-*d*_6_) δ: 10.67 (s, 1 H), 7.83 (m, 1 H), 7.15 (m, 2 H), 5.64 (s, 1 H), 4.19 (q, *J* 7.1 Hz, 2 H), 1.26 (t, *J* 7.1 Hz, 3 H) ppm.


**General procedure for compounds 10**
**o-q.**


DMAD (4.1 mmol, 0.5 mL) was added to a stirring suspension of a corresponding *o*-aminophenol **9a**,**g**,**h** (4.1 mmol) in methanol (5 mL). The reaction mixture was stirred at room temperature overnight. The formed yellow precipitate was filtered off, washed with methanol (15 mL) and recrystallized from ethanol (50 mL) to give the corresponding compound **10o**-**q**.

*Methyl (Z)-2-(2-oxo-2H-benzo[b][1,4]oxazin-3(4H)-ylidene)acetate (**10o**)* [42]. Yield: 0.68 g (76%); yellow solid; mp 163–163 °C (ethanol) (lit. 163–165 °C [42]). ^1^H NMR (400 MHz, DMSO-*d*_6_) δ: 10.66 (s, 1 H), 7.50 (m, 1 H), 7.17 (m, 2 H), 7.04 (m, 1 H), 5.64 (s, 1 H), 3.72 (s, 3 H) ppm.

*Methyl (Z)-2-(6-chloro-2-oxo-2H-benzo[b][1,4]oxazin-3(4H)-ylidene)acetate (**10p**)* [57]. Yield: 0.74 g (71%); yellow solid; mp 163–166 °C (ethanol) (lit. 164–166 °C [57]). ^1^H NMR (400 MHz, DMSO-*d*_6_) δ: 10.66 (s, 1 H), 7.71 (m, 1 H), 7.20 (m, 1 H), 7.04 (m, 1 H), 5.67 (s, 1 H), 3.72 (s, 3 H) ppm.

*Methyl (Z)-2-(6-bromo-2-oxo-2H-benzo[b][1,4]oxazin-3(4H)-ylidene)acetate (**10q**).* Yield: 0.82 g (67%); yellow solid; mp 191–193 °C (ethanol). ^1^H NMR (DMSO-*d*_6_, 400 MHz) δ: 10.64 (s, 1 H), 7.84 (m, 1 H), 7.15 (m, 2 H), 5.66 (s, 1 H), 3.72 (s, 3 H) ppm. ^13^C NMR (DMSO-*d*_6_, 100 MHz) δ: 168.0, 155.4, 139.1, 137.9, 126.2, 124.5, 118.3, 117.9, 116.5, 89.6, 51.0 ppm. IR (mineral oil): 3200, 1782 cm^−1^. Anal. Calcd (%) for C_11_H_8_BrNO_4_: C 44.32; H 2.71; N 4.70. Found: C 44.09; H 2.80; N 4.68.

*(Z)-2-(2-Oxo-2-phenylethylidene)-1,5-dihydrobenzo[e][1,4]oxazepin-3(2H)-one (**11**)* [58]. A mixture of benzoylpyruvic acid **3a** (26 mmol, 5.0 g), *o*-aminobenzyl alcohol (26 mmol, 3.2 g), benzene (100 mL), and acetic acid (52 mmol, 3 mL) was refluxed in a flask equipped with a Dean–Stark apparatus for 6 h. Then, the solvent was removed, and the residue was ground with ethanol. The formed precipitate of compound **11** was filtered off and recrystallized from ethanol (50 mL). Yield: 2.98 g (41%); yellow solid; mp 164–166 °C (ethanol) (lit. 164–166 °C [58]). ^1^H NMR (400 MHz, DMSO-*d*_6_) δ: 12.83 (s, 1 H), 8.01 (m, 2 H), 7.62 (m, 1 H), 7.54 (m, 3 H), 7.44 (m, 2 H), 7.23 (m, 1 H), 6.56 (s, 1 H), 5.35 (s, 2 H) ppm.

*2-Hydroxy-2-(2-oxo-2-phenylethyl)-2H-benzo[b][1,4]thiazin-3(4H)-one (**13a**)* [55]. To a stirring solution of 5-phenylfuran-2,3-dione **12a** (1 mmol, 174 mg) in 1,4-dioxane (3 mL), *o*-aminothiophenol (1.05 mmol, 110 μL) was added. The mixture was stirred at room temperature overnight. Then, the solvent was removed under vacuum. The solid residue was stirred with toluene (3 mL) for 1 h. Then, the precipitate was filtered off and washed with acetone (3 mL) to afford the compound **13a**. Yield: 0.23 g (76%); pale yellow solid; mp 183–185 °C (acetone) (lit. 183–183 °C [55]). ^1^H NMR (400 MHz, DMSO-*d*_6_) δ: 10.64 (s, 1 H), 7.97 (m, 2 H), 7.64 (m, 1 H), 7.53 (m, 2 H), 7.27 (m, 1 H), 7.19 (m, 1 H), 7.05 (m, 1 H), 6.98 (m, 1 H), 6.90 (s, 1 H), 3.74 (d, *J* 16.1 Hz, 1 H), 3.66 (d, *J* 16.3 Hz, 1 H) ppm.

*(Z)-2-(2-Oxo-2-(p-tolyl)ethylidene)-2H-benzo[b][1,4]thiazin-3(4H)-one (**14**).* To a suspension of 2-hydroxy-2-(2-oxo-2-(*p*-tolyl)ethyl)-2*H*-benzo[*b*][1,4]thiazin-3(4*H*)-one **13b** (1.6 mmol, 0.5 g) in benzene (5 mL), oxalyl chloride (3.2 mmol, 0.3 mL) was added. The mixture was refluxed for 2 h. Then, the reaction mixture was cooled to room temperature. The formed precipitate was filtered off, washed with ethanol (2 mL) and recrystallized from benzene (10 mL) to result in compound **14.** Yield: 0.47 g (88%, a benzene hemisolvate); dark yellow solid; mp 262–263 °C (benzene). ^1^H NMR (DMSO-*d*_6_, 400 MHz) δ: 11.63 (s, 1 H), 8.24 (s, 1 H), 7.94 (m, 2 H), 7.48 (m, 1 H), 7.39 (m, 2 H), 7.29 (m, 1 H), 7.18 (m, 1 H), 7.13 (m, 1 H), 2.40 (s, 3 H) ppm. ^13^C NMR (DMSO-*d*_6_, 100 MHz) δ: 187.7, 154.7, 143.5, 141.7, 134.9, 133.0, 129.5, 127.9, 127.4, 125.4, 123.3, 116.8, 116.8, 115.5, 21.0 ppm. IR (mineral oil): 3180, 1667 cm^−1^. Anal. Calcd (%) for 2C_17_H_13_NO_2_S · C_6_H_6_: C 71.83; H 4.82; N 4.19. Found: C 72.09; H 4.80; N 4.27. Crystal structure of compound **14** was deposited at the Cambridge Crystallographic Data Centre with the deposition number CCDC 2486573.

*(Z)-3-(2-(4-Chlorophenyl)-2-oxoethylidene)-4-methyl-3,4-dihydroquinoxalin-2(1H)-one (**15**)* [54]. To a stirred suspension of 5-(*p*-tolyl)furan-2,3-dione **12b** (5 mmol, 0.94 g) in toluene (5 mL) *N*^1^-methylbenzene-1,2-diamine **5c** (5 mmol, 568 μL) was added. The mixture was stirred at room temperature for 12 h. The formed precipitate was filtered off, washed with hot toluene (20 mL) and ethanol (10 mL) to afford the compound **15**. Yield: 0.95 g (91%); orange solid; mp 221–223 °C (toluene) (lit. 221–233 °C [54]). ^1^H NMR (400 MHz, DMSO-*d*_6_) δ: 11.82 (s, 1 H), 7.96 (m, 2 H), 7.57 (m, 2 H), 7.36 (m, 1 H), 7.19 (m, 1 H), 7.16 (m, 2 H), 6.96 (s, 1 H), 3.37 (s, 3 H) ppm.

*Diethyl 2-((2-acetamidophenyl)amino)fumarate (**16**)* [42]. Diethyl oxaloacetate **2p** (4.0 mmol, 0.75 g) was added to a suspension of *N*-(2-aminophenyl)acetamide **5d** (4.0 mmol, 0.6 g) in ethanol (10 mL). The reaction mixture was refluxed for 2 h, cooled to ambient temperature. The formed yellow precipitate was filtered off, washed with ethanol (5 mL) and recrystallized from ethanol (10 mL) to give compound **16**. Yield: 0.85 g (66%); yellow solid; mp 99–101 °C (ethanol) (lit. 99–101 °C [42]). ^1^H NMR (400 MHz, DMSO-*d*_6_) δ: 9.88 (s, 1 H), 9.40 (s, 1 H), 7.26 (m, 1 H), 7.13 (m, 2 H), 6.81 (m, 1 H), 5.22 (s, 1 H), 4.12 (q, *J* 7.1 Hz, 2 H), 4.01 (q, *J* 7.1 Hz, 2 H), 2.08 (s, 3 H), 1.23 (t, *J* 7.1 Hz, 3 H), 0.96 (t, *J* 7.1 Hz, 3 H) ppm.

### 2.2. Biology

#### 2.2.1. Screening of Substances in 96-Well Plates and Evaluation of Compounds in 50 mL Flasks

The screening of test compounds was conducted in 96-well microplates, followed by validation of the most promising candidates in 50 mL Erlenmeyer flasks. Experimental procedures were based on our previously established protocol [23], with modifications as detailed below.

*Chlorella vulgaris* strain IMBR-19 (The A.O. Kovalevsky Institute of Biology of the Southern Seas, RAS, Sevastopol, Russia) was cultivated axenically in BG-11 medium [59] supplemented with 1% (*v*/*v*) DMSO. Test compounds were dissolved in DMSO and introduced to cultures at final concentrations of 0.1, 1.0, and 10.0 μmol/L for microplate assays, and at 0.1, 1.0, 10.0, and 100.0 μmol/L for flask experiments. Glucose (2 g/L) served as a positive control, while negative controls contained DMSO alone.

For microplate assays, each well was loaded with 300 μL of cell suspension at an initial density of 5 × 10^4^ cells per well. Plates were sealed with sterile gas-permeable membranes and incubated for five days at 28 °C with orbital shaking at 150 rpm in a humidified chamber. Cultures were maintained under a 12 h light/12 h dark photoperiod, with an incident light intensity of 100 μmol photons m^−2^ s^−1^ provided by an array of white LEDs positioned beneath the plates to ensure uniform bottom-up illumination. A dedicated cooling system prevented thermal transfer from the light source to the cultures.

In flask-based assays, cultures (30 mL) were grown under identical medium composition and environmental conditions as described above.

Cell density was estimated by measuring optical density at 750 nm (OD750) using a microplate reader [60]. When microalgae were cultivated in flasks, the lower number of samples allowed for more accurate direct cell counts using a hemocytometer, rather than measuring OD750. The cell concentration data were used both to assess the effects of the compounds on C. vulgaris growth and to normalize metabolite content to cell concentration. Photosynthetic pigment concentrations (chlorophylls and carotenoids) were determined spectrophotometrically according to established protocols [61,62]. Total protein content was quantified via a modified Bradford assay [63], employing bovine serum albumin (BSA) as the standard for calibration. For protein determination, cells were subjected to acid hydrolysis prior to analysis. Total carbohydrate content was assessed using a modified anthrone colorimetric method [64], with glucose as the calibration standard.

#### 2.2.2. Data Analysis

For the screening assays, each tested compound concentration was assessed in duplicate wells, while six wells were designated for both the negative and positive control groups. Compounds were considered for further evaluation if their average OD750 value exceeded the mean OD750 of the negative control by more than three standard deviations. To ensure that the microplate assay exhibited adequate discriminatory power, the Z′ factor was calculated for each plate using the following formula:$$Z^{\prime }=1-\frac{3\times ({SD}_{pos}-{SD}_{neg})}{m_{pos}-m_{neg}}$$
where m_pos_ and m_neg_ represent the mean OD750 values of the positive and negative controls, respectively, and SD_pos_ and SD_neg_ are their corresponding standard deviations. Only plates with a Z′ factor greater than 0.5 were considered valid [65].

In the in-depth analysis of compounds bioactivity, three independently cultured flasks were used as biological replicates. Positive controls, however, utilized biological replicates derived from a single culture flask for each condition. Protein measurements were carried out using three technical replicates (n = 3) per biological replicate, resulting in a total of nine data points for each condition. For pigment and carbohydrate assessments, two technical replicates (n = 2) were taken from each biological replicate, providing six data points per condition. Statistical analyses across groups were performed using nested one-way ANOVA.

Sample size calculations for microplate screening experiments were conducted using G*Power 3.1.9.7. Parameters were set to achieve a statistical power of 90% (corresponding to a Type II error rate of 10%) and a Type I error rate of 5%. The number of wells in the control group was fixed at six, assuming equal standard deviations across all groups. Calculations were based on the independent samples two-tailed *t*-test. Under these assumptions, a minimum of two wells per treatment group was determined to be sufficient.

The activity threshold was defined as the mean value plus three standard deviations (mean + 3 × SD), which is based on data from paper [28], where a relatively low hit rate (compounds inducing a ≥50% increase in cell concentration) was observed (<0.1%). Due to the limited size of our compound library, we adopted this relatively non-stringent threshold to facilitate the detection of statistically significant differences between control and treatment groups, thereby increasing the likelihood of identifying potential hits. However, this lower threshold may result in the selection of compounds that induce statistically significant, but not necessarily biologically meaningful, increases in cell concentration.

## 3. Results and Discussion

### 3.1. Synthesis of Acylpyruvates and Their Derivatives

To the best of our knowledge, there are no clear SAR established for acylpyruvates and their heterocyclic derivatives in the context of growth regulation in microalgae. Therefore, we employed a rationally diverse design strategy, incorporating both systematic and randomized substitution patterns. While focusing primarily on six-membered nitrogen-, oxygen-, and sulfur-containing heterocycles (pyran **4**, piperazine **8**, 1,4-benzoxazines **10**, 1,4-benzothiazines **7**, **13**, **14**, and quinoxalines **6**, **15**), we also included one seven-membered heterocyclic derivative **11** and one open-chain derivative **16** to broaden the structural scope and evaluate potential scaffold-dependent effects.

A library of 55 compounds, representing 12 chemotypes (including acylpyruvic acids **3**, their esters **2**, heterocyclic derivatives **4**, **6**–**8**, **10**, **11**, **13**–**15**, and one open-chain analog **16**), was synthesized using previously reported protocols (Figure 2, Figure 3 and Figure 4) [38,42,43,44,50,52,54,55,56,58,66,67].

In brief, methyl esters of acylpyruvic acids **2a**-**m**,**o**,**q**-**s** were prepared via the Claisen condensation of diethyl oxalate with methyl ketones **1a**-**m**,**o**,**q**-**s** in the presence of sodium methoxide (Figure 2) [38,43,66]. Diethyl oxaloacetate **2p** was prepared by the Claisen condensation of diethyl oxalate with ethyl acetate **1p** in the presence of sodium and sodium ethoxide (Figure 2) [42]. Diethyl (2*Z*,5*Z*)-2,6-dihydroxy-4-oxohepta-2,5-dienedioate **2n** was prepared by the Claisen condensation of two equivalents of diethyl oxalate with acetone **1n** in the presence of two equivalents of sodium ethoxide (Figure 2) [44]. Acylpyruvic acids **3a**-**l** were prepared by alkaline hydrolysis of the corresponding methyl esters **2a**-**j**,**q**,**s** (Figure 2) [67]. Chelidonic acid **4** was prepared by acidic hydrolysis of diethyl (2*Z*,5*Z*)-2,6-dihydroxy-4-oxohepta-2,5-dienedioate **2n** (Figure 2) [44]. *N*^4^-Unsubstituted quinoxalin-2-ones **6a**-**g** were prepared by the reaction of esters of acylpyruvic acids **2a**,**c**,**p**,**r** with *o*-phenylenediamines **5a**-**c** (Figure 2) [50,52,54]. The reaction of acylpyruvic acids **3h**,**j**,**q**,**s** with *o*-aminophenol in the presence of dicyclohexylcarbodiimide (DCC) afforded 1,4-benzothiazin-2-ones **7a**-**d** (Figure 2) [55]. The reaction of 4-toluoylpyruvic acid **3j** with 1,2-ethylenediamine afforded piperazin-2-one **8** (Piron) (Figure 2) [43,56]. 1,4-Benzoxazin-2-ones **10a**-**n** were prepared by the reaction of esters of acylpyruvic acids **2a**,**i**,**o**,**p** with *o*-aminophenols **9a**-**h**; methoxycarbonyl-bearing 1,4-benzoxazin-2-ones **10o**-**q** were prepared by the reaction of dimethyl acetylenedicarboxylate (DMAD) with *o*-aminophenols **9a**,**g**,**h** (Figure 3) [42]. The reaction of benzoylpyruvic acid **3a** with (2-aminophenyl)methanol afforded benzo[*e*][1,4]oxazepin-3-one **11** (Figure 2) [58]. Acylpyruvic acids **3a**,**f**,**j** were dehydrated by SOCl_2_ to result in 5-arylfuran-2,3-diones **12a**-**c** (Figure 2) [54]. The reaction of 5-arylfuran-2,3-diones **12a**,**c** with *o*-aminothiophenol afforded 2-hydroxy-1,4-benzothiazin-3-ones **13a**,**b** (Figure 2) [55]. 2-Hydroxy-1,4-benzothiazin-3-one **13b** dehydrated by (COCl)_2_ to result in 1,4-benzothiazin-3-one **14** (Figure 2). The reaction of 5-(4-chlorophenyl)furan-2,3-dione **12b** with *N*^1^-methylbenzene-1,2-diamine **5c** afforded 2-hydroxy-1,4-benzothiazin-3-one **15** (Figure 2) [54]. The reaction of diethyl oxaloacetate **2p** with *N*-(2-aminophenyl)acetamide **5d** afforded open-chain derivative **16** (Figure 4) [42].

### 3.2. Biology

The biotechnological potential of the 55 synthesized pyruvate derivatives was evaluated by assessing their effects on *C. vulgaris* cultures. The compounds tested included derivatives **2a**-**f**,**j**-**n**, **3a**-**i**, **4**, **6a**-**g**, **7a**-**d**, **8**, **10a**-**q**, **11**, **13a**, **14**, **15**, and **16** (Table 1, entries 1–55). Stock solutions of the compounds were prepared in DMSO and introduced into the cultures at final concentrations ranging from 0.1 to 10 μmol/L. Cultures were grown in 96-well plates, with 1% DMSO serving as a negative control and 2 g/L glucose as a positive control. After five days of cultivation, the concentration of *C. vulgaris* cells was determined by measuring the optical density at 750 nm (OD750). Compounds were considered active if they increased OD750 compared to the negative control (defined as OD750 exceeding the mean value of the negative control plus three standard deviations) at two or more tested concentrations. For 15 of the tested compounds (compounds **2a**-**d**,**f**,**j**,**k**, **3a**,**b**,**d**-**i**), solubility in DMSO varied significantly, necessitating two distinct stock preparation protocols. In one approach, the compounds were mixed with DMSO and added to the wells after 1–2 h; in the other, the compounds were incubated overnight at 37 °C prior to addition. Consequently, two OD750 values were obtained for these compounds based on the different solubilization regimes. A compound was selected for further detailed investigation if it satisfied the activity criterion under at least one of the solubilization conditions.

Screening of the 55-compound library, representing 12 distinct chemotypes, identified six compounds that met the activity criterion and significantly enhanced *C. vulgaris* growth (Table 1, entries 2, 4, 13, 16, 21, 31). The active compounds belonged to only four of the twelve tested chemotypes: methyl esters of (het)aroylpyruvic acids **2b**,**d**, free (het)aroylpyruvic acids **3b**,**e**, chelidonic acid **4**, and 1,4-benzothiazin-2-one **7c**. Four of the six active compounds are direct acylpyruvate derivatives **2**, **3**, identifying underivatized acylpyruvates as the most promising class of compounds for further investigations.

Notably, a significant number of 1,4-benzoxazin-2-ones **10** (specifically, compounds **10a**,**b**,**h**,**k**-**q**), intensively studied by many research groups as potent source of new drugs [42,57,68,69,70,71,72], exhibited a potent growth inhibitory effect. At the highest tested concentration, these derivatives caused a near-complete suppression of *C. vulgaris* growth (Table 1, entries 34, 35, 41, 44–50). The inhibitory effect was dose-dependent for most compounds in this series, diminishing at lower concentrations. This growth suppression suggests that the 1,4-benzoxazin-2-one **10** scaffold can confer inherent toxicity towards microalgae, making these compounds potent algicides. However, we propose that the toxicity of 1,4-benzoxazin-2-ones **10** towards *C. vulgaris* can be explained by their hydrolysis in an aqueous medium to form *o*-aminophenols **9** (Figure 5), which are known to be toxic to microalgae [73]. To test this hypothesis, we evaluated the effects of *o*-aminophenol **9a** on *C. vulgaris* cultures under the same 96-well plates micro assay conditions used for the 1,4-benzoxazin-2-ones **10**. Consistent with our proposal, *o*-aminophenol **9a** also demonstrated significant toxicity (Table 1, entry 56). Therefore, the observed algicidal activity may be a result of either the intrinsic properties of the 1,4-benzoxazin-2-one scaffold itself or its hydrolysis products, or a combination of both mechanisms. Furthermore, the discovery of the potent algicidal activity associated with the 1,4-benzoxazin-2-one scaffold (**10a**,**b**,**h**,**k**-**q**) warrants a discussion on its application safety. If developed as biocides, a comprehensive assessment of their environmental impact, including cytotoxicity to other aquatic life and biodegradability, would be a necessary next step.

Then, for a detailed investigation of their bioactivity, compounds **2b**,**d**, **3b**,**e**, **4**, **7c** were further evaluated for their effects on cell concentration, as well as on pigment, protein, polysaccharide, and neutral lipid content (Table 2, Table 3, Table 4, Table 5, Table 6 and Table 7). For these experiments, the selected compounds were added to *C. vulgaris* cultures at concentrations ranging from 0.1 to 100 μmol/L. Cultures were grown in 50 mL flasks, and after five days of cultivation, samples were analyzed using standard biochemical methods with minor modifications (described in detail in the Section 2.2).

Interestingly, despite their promising performance in microplate-based screenings, the tested compounds did not exhibit a statistically significant effect on cell concentration in flask cultures when compared to the negative control (Table 2, Table 3, Table 4, Table 5, Table 6 and Table 7). The sole exception was compound **3b**, which resulted in an 11.5% increase in cell concentration. This discrepancy between the microplate and flask culture systems may be explained by differences in mixing dynamics and oxygen availability, which could influence the presence or concentration of specific signaling or regulatory molecules [74] that serve as targets for the tested compounds.

Notably, several compounds demonstrated the ability to enhance the intracellular content of specific metabolites: compound **2b** increased chlorophyll levels (17% growth), compound **3b** elevated carotenoid content (40% growth), and compound **2d** promoted the accumulation of neutral lipids (44% growth). However, these effects were frequently accompanied by a reduction in overall cell concentration.

Two of the six active compounds are *p*-ethoxybenzoylpyruvates **2b** and **3b**, which may be highlighting the *p*-ethoxybenzoyl substituent incorporated in a pyruvate core as a promising structural feature for the design of more potent analogs in future SAR studies.

However, to ascertain whether the observed effects on *C. vulgaris* are indeed attributed to the *p*-ethoxybenzoylpyruvate scaffold—and not to its potential hydrolysis products in the aqueous medium—we evaluated *p*-ethoxyacetophenone **1b** and *p*-ethoxybenzoic acid **17**, which are the primary degradation products expected under these conditions [4,75] (Figure 6). This approach was particularly relevant as similar benzoic acids and acetophenone derivatives have been reported to exhibit stimulating effects on algae [31,32].

To evaluate the potential role of the hydrolysis products, we measured the pigment content in *C. vulgaris* cultures supplemented with compounds **1b** and **17** at concentrations of 10 and 100 μmol/L (Table 8). At 100 μmol/L, compound **17** increased the total chlorophyll content by 11%, while both compounds **1b** and **17** at this concentration enhanced the carotenoid content by 19–20%. However, these effects were not statistically significant (*p* < 0.05, nested ANOVA). Furthermore, and in contrast to the parent compounds **2b** and **3b**, neither compound **1b** nor compound **17** significantly altered the cell density. These findings suggest that the stimulatory effects of compounds **2b** and **3b** on *C. vulgaris* are likely mediated by the intact *p*-ethoxybenzoylpyruvate scaffold itself, rather than by its hydrolysis products.

Overall, our experimental data demonstrate that compounds **2b**,**d**, **3b** can act as metabolic modulators in *C. vulgaris* cultures, significantly enhancing the intracellular content of specific metabolites under the controlled conditions of this study. It is important to note that the most pronounced stimulatory effects for compounds such as **2b**,**d**, **3b**,**e**, **4**, and **7c** were observed in the initial microplate screening at concentrations ranging from 0.1 to 10 μmol/L, an environment characterized by limited mixing and moderate aeration. This observed bioactivity positions them as promising candidates for further development. Future research should focus on optimizing their application, including testing in combination with other growth stimulators and under industrially relevant cultivation systems to validate their potential for simultaneously increasing both cell density and metabolite yield.

## 4. Conclusions

In conclusion, our study establishes the first SAR for acylpyruvate-derived compounds as growth stimulants for the biotechnologically relevant microalga *C. vulgaris*. Screening of a 55-compound library identified six promising compounds (methyl esters of (het)aroylpyruvic acids **2b**,**d**, free (het)aroylpyruvic acids **3b**,**e**, chelidonic acid **4**, and 1,4-benzothiazin-2-one **7c**), and the underivatized acylpyruvate pattern emerged as the most active scaffold. While a subset of heterocyclic derivatives—1,4-benzoxazin-2-one **10**—exhibited growth inhibitory effects, the primary focus of our research was on stimulators that enhanced biomass in initial microplate screenings. Most notably, under scaled-up culture conditions, the active compounds **2b**,**d**, **3b**,**e**, **4**, and **7c** demonstrated a valuable ability to act as metabolic modulators, specifically enhancing the intracellular accumulation of high-value metabolites—including chlorophylls, carotenoids, and neutral lipids—by up to 44%, even in cases where the effect on overall cell density was limited. These findings position acylpyruvate derivatives as highly promising and versatile candidates for developing novel biostimulants to increase the productivity of microalgal cultivation systems.

## Data Availability

The presented data are available in this article/Appendix A.

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
