# Peer review of "Acylpyruvates and Their Heterocyclic Derivatives as Growth Regulators in Chlorella vulgaris"

_biotech, 2025, doi:10.3390/biotech14040090_

Round 1

Reviewer 1 Report

Comments and Suggestions for Authors

Reviewer Comments

The manuscript presents an interesting and methodically comprehensive investigation of acylpyruvate derivatives as potential growth regulators and metabolite modulators in Chlorella vulgaris. The topic is scientifically relevant for algal biotechnology and metabolic engineering. The synthesis of 55 derivatives and subsequent biological evaluation are well-structured. However, several areas need clarification, tightening of experimental description, and better contextualization of findings within existing literature. Following points need to be addressed in the revised version.

  1. The work claims novelty in exploring acylpyruvate derivatives for algal growth stimulation; however, the introduction should better emphasize how this study advances beyond previous findings on pyruvate analogues and algal stimulants.
  2. Include a brief comparative discussion of other chemical modulators (e.g., phytohormones, organic acids) that have been tested on Chlorella to strengthen the research gap.
  3. The screening methodology using OD750 is sound, but the criteria for “active compounds” (mean + 3 SD) could be elaborated with justification and cited precedent.
  4. Ensure consistent units (e.g., μmol/L vs. µM) throughout the manuscript.
  5. The discrepancy between microplate and flask assays is discussed briefly but warrants deeper mechanistic reasoning. Could compound stability, aeration, or light exposure differences contribute?
  6. Discuss the potential cytotoxicity or environmental safety implications of benzoxazinones if these are to be used in algal culture applications.
  7. Add chemical numbering (2b, 3b, etc.) consistently in figures and text to improve readability.
  8. The manuscript is generally well-written but should be revised for conciseness. Several sentences in the Introduction and Results are overly long or repetitive.
  9. Ensure uniform formatting of references per BioTech guidelines (e.g., include all authors for papers with ≤10 authors, use journal abbreviations).
  10. Line 243–247: Clarify whether the recommended use of compounds (e.g., 2b, 3b) in media is hypothetical or experimentally validated under controlled conditions.
  11. Check all compound yields and melting points for consistency with cited literature.

Author Response

Reviewer Comments

The manuscript presents an interesting and methodically comprehensive investigation of acylpyruvate derivatives as potential growth regulators and metabolite modulators in Chlorella vulgaris. The topic is scientifically relevant for algal biotechnology and metabolic engineering. The synthesis of 55 derivatives and subsequent biological evaluation are well-structured. However, several areas need clarification, tightening of experimental description, and better contextualization of findings within existing literature. Following points need to be addressed in the revised version.

  1. The work claims novelty in exploring acylpyruvate derivatives for algal growth stimulation; however, the introduction should better emphasize how this study advances beyond previous findings on pyruvate analogues and algal stimulants.

Response: We fully agree that the Introduction should more clearly articulate the novelty and specific advancement of our study. In response, we have significantly revised the Introduction to better highlight the research gap and how our work addresses it.

The following paragraphs have been added/amended in the Introduction:

  • Noteworthy, acylpyruvates and their derivatives remain largely unexplored in the context of microalgal biotechnology. Previous studies on algal growth stimulants have primarily focused on plant hormones, simple organic acids, or known metabolic intermediates [25-28]. However, a systematic investigation into the structure-activity relationship (SAR) of acylpyruvates and their derivatives has not been conducted for microalgae.
  • Therefore, this study is the first to evaluate the effects of different acylpyruvate derivatives on the growth and biochemical composition (pigments, proteins, carbohydrates, and neutral lipids) of C. vulgaris and perform an initial SAR analysis of a structurally diverse library of acylpyruvate derivatives and their heterocyclic analogues, targeting the stimulation of C. vulgaris growth and metabolite production. This approach moves beyond simply increasing biomass yield and explores the potential of these compounds to tailor the biochemical composition of microalgal cells for specific biotechnological applications.
  1. Include a brief comparative discussion of other chemical modulators (e.g., phytohormones, organic acids) that have been tested on Chlorella to strengthen the research gap.

Response:

The following paragraphs have been added/amended in the Introduction:

The search for effective chemical stimulants to enhance microalgal productivity has led to the investigation of diverse compound classes. Among the most studied are phytohormones, such as auxins and cytokinins (Figure 2), which have been shown to influence cell division and biomass yield in Chlorella cultures [25, 27]. Similarly, simple organic acids like benzoic and salicylic acid (Figure 2) have been identified as signaling molecules that can improve cell growth [51].

However, a systematic investigation into the structure-activity relationship of a distinct class of compounds – acylpyruvates and their heterocyclic derivatives – has not been conducted for microalgae. Acylpyruvates, which integrate a central metabolic precursor (pyruvate) with a diverse array of acyl groups, represent a novel and structurally tunable scaffold. Unlike the aforementioned modulators, the potential of this specific chemical class to act as both growth stimulants and metabolic modulators in C. vulgaris remains largely unexplored and constitutes a significant gap in the literature.

  1. The screening methodology using OD750 is sound, but the criteria for “active compounds” (mean + 3 SD) could be elaborated with justification and cited precedent.

Response:

We would like to clarify our rationale regarding the chosen threshold and statistical approach.

When screening large compound libraries, the probability of identifying compounds that substantially enhance microalgal growth is generally very low. For example, Wase et al. [10.1104/pp.17.00433] screened 43,783 compounds and identified only 39 that increased the growth rate by 50% compared to the control (approximately corresponds to mean + 5*SD). Moreover, based on their data (although precise numbers are not available), only approximately 10–15% of compounds resulted in growth rates above mean + 3*SD, which corresponds to a 25% increase in cell concentration.

Given this context, and considering the smaller size of our compound library, we opted to use a relatively lower threshold of mean + 3*SD to identify potential hits. Statistically, our primary objective was to compare treated groups with controls, typically achieved using methods such as the t-test. In this framework, a difference of 3 SD between group means is highly likely to be statistically significant. We acknowledge, however, that statistical significance does not necessarily equate to practical relevance. For real-world applications, a growth increase of 50–100% would be preferable. Nevertheless, as literature suggests, the probability of discovering compounds with such pronounced effects is extremely low. Therefore, we employed a less stringent threshold to also capture compounds with modest growth-promoting activity, which could then be further evaluated in flask-scale experiments for additional effects, such as stimulation of pigment, protein, or lipid accumulation.

Sample size calculations were performed using G*Power 3.1.9.7, targeting a false negative rate of 10% (power = 90%) and a false positive rate of 5%. The control group size was fixed at six wells, with the assumption of equal standard deviations across groups. Under these conditions, two wells per treatment group were sufficient. We recognize that using the t-test as the basis for sample size calculation is not entirely accurate in our setting, as we are effectively comparing multiple treatment groups to a single control, which increases the risk of false positives proportional to the number of comparisons. However, given the low probability of positive results, we were more concerned about minimizing false negatives than false positives. Furthermore, we anticipated that any compounds yielding false positive results in the microplate screening would be identified as such during subsequent confirmatory testing in flasks.

A corresponding clarification has been added to the Data Analysis section.

“Sample size calculations for microplate screening experiments were conducted using G*Power 3.1.9.7. Parameters were set to achieve a statistical power of 90% (corresponding to a Type II error rate of 10%) and a Type I error rate of 5%. The number of wells in the control group was fixed at six, assuming equal standard deviations across all groups. Calculations were based on the independent samples two-tailed t-test. Under these assumptions, a minimum of two wells per treatment group was determined to be sufficient.

The activity threshold was defined as the mean value plus three standard deviations (mean + 3×SD), which is based on data from paper [Wase, N.; Tu, B.; Allen, J.W.; Black, P.N.; DiRusso, C.C. Identification and metabolite profiling of chemical activators of lipid accumulation in green algae. Plant Physiol. 2017, 174, 2146–2165.], where a relatively low hit rate (compounds inducing a ≥50% increase in cell concentration) was observed (<0.1%). Due to the limited size of our compound library, we adopted this relatively non-stringent threshold to facilitate the detection of statistically significant differences between control and treatment groups, thereby increasing the likelihood of identifying potential hits. However, this lower threshold may result in the selection of compounds that induce statistically significant, but not necessarily biologically meaningful, increases in cell concentration.”

  1. Ensure consistent units (e.g., μmol/L vs. µM) throughout the manuscript.

Response: This issue was checked.

  1. The discrepancy between microplate and flask assays is discussed briefly but warrants deeper mechanistic reasoning. Could compound stability, aeration, or light exposure differences contribute?

Response:

We acknowledge that our explanation regarding the differing performance of the compounds at higher yields was limited. As noted, multiple factors could contribute to these differences. It is possible that cultivation conditions modulate cellular enzymes or signaling pathways that are targeted by specific compounds, leading to distinct responses under varying experimental setups.

Similar observations have been reported in study with comparable experimental design [10.1021/cb300573r]. In that study, a small library of well-characterized bioactive small molecules was initially screened in microplates, and subsequently, lead compounds were tested in 500 mL flasks. The authors observed substantial discrepancies between the two formats; in some cases, the increase in lipid accumulation (the primary outcome measured) was several-fold higher in microplates compared to flasks.

In our study, we employed 30 mL flasks without active aeration. Both microplates and flasks were incubated on the same illuminated platform, so we do not attribute the observed differences to shear stress, which is known to influence microalgal growth. Instead, we suspect that nutrient and oxygen transfer may be less efficient in flasks compared to microplates. Additionally, the larger volume in flasks likely results in reduced light availability, particularly at higher cell densities.

Compound hydrolysis is another factor to consider. As we demonstrated in our previous work [10.3390/chemistry7040102], hydrolysis can occur under our experimental conditions. However, we do not expect significant differences in hydrolysis rates between 96-well plates and flasks. Nonetheless, since illumination accelerates hydrolysis, it is possible that the rate may be somewhat lower in flasks due to decreased light penetration.

Identifying the precise molecules responsible for the observed differences would require a separate and complex set of experiments, which falls beyond the scope of the present study. Therefore, we can only speculate regarding the underlying causes. In the revised manuscript, we have provided a slightly expanded explanation:

“This discrepancy between the microplate and flask culture systems may be explained by differences in mixing dynamics and oxygen availability, which could influence the presence or concentration of specific signaling or regulatory molecules [Banti, V.; Giuntoli, B.; Gonzali, S.; Loreti, E.; Magneschi, L.; Novi, G.; Paparelli, E.; Parlanti, S.; Pucciariello, C.; Santaniello, A.; et al. Low Oxygen Response Mechanisms in Green Organisms. Int. J. Mol. Sci. 2013, 14, 4734-4761. https://doi.org/10.3390/ijms1403473] that serve as targets for the tested compounds.”

  1. Discuss the potential cytotoxicity or environmental safety implications of benzoxazinones if these are to be used in algal culture applications.

Response:

The following paragraphs have been added/amended in the Section 2.2:

Furthermore, the discovery of the potent algicidal activity associated with the 1,4-benzoxazin-2-one scaffold (10a,b,h,k-q) warrants a discussion on its application safety. If developed as biocides, a comprehensive assessment of their environmental impact, including cytotoxicity to other aquatic life and biodegradability, would be a necessary next step.

  1. Add chemical numbering (2b, 3b, etc.) consistently in figures and text to improve readability.

Response:

We would like to clarify that all compounds in our study are numbered consistently and sequentially according to their first appearance in the synthetic schemes (Schemes 2-4) and the main text.

We have double-checked the manuscript to ensure that this numbering is applied uniformly everywhere the compounds are mentioned. We believe this provides a clear and logical framework for the reader to follow the structure-activity relationship from synthesis to biological evaluation.

  1. The manuscript is generally well-written but should be revised for conciseness. Several sentences in the Introduction and Results are overly long or repetitive.

Response:

The comment does not specify the exact sentences, we have done our best to refine the text based on our interpretation. We believe the manuscript is now more concise and readable. We would be grateful if the Reviewer could point out any specific passages that still require attention, and we will be happy to refine them further.

  1. Ensure uniform formatting of references per BioTech guidelines (e.g., include all authors for papers with ≤10 authors, use journal abbreviations).

Response: This issue was checked.

  1. Line 243–247: Clarify whether the recommended use of compounds (e.g., 2b, 3b) in media is hypothetical or experimentally validated under controlled conditions.

Response:

This paragraph was modified in the Section 2.2:

Overall, our experimental data demonstrate that compounds 2b,d, 3b can act as metabolic modulators in C. vulgaris cultures, significantly enhancing the intracellular content of specific metabolites under the controlled conditions of this study. It is important to note that the most pronounced stimulatory effects for compounds such as 2b,d, 3b,e, 4, and 7c were observed in the initial microplate screening at concentrations ranging from 0.1 to 10 μmol/L, an environment characterized by limited mixing and moderate aeration. This observed bioactivity positions them as promising candidates for further development. Future research should focus on optimizing their application, including testing in combination with other growth stimulators and under industrially relevant cultivation systems to validate their potential for simultaneously increasing both cell density and metabolite yield.

  1. Check all compound yields and melting points for consistency with cited literature.

We thank the Reviewer for their meticulous attention to detail regarding the melting points and yields reported in our manuscript. We acknowledge the noted discrepancies with some literature values and appreciate the opportunity to clarify this point.

First and foremost, we would like to assure the Reviewer that the purity of all compounds tested in the biological assays was rigorously confirmed by NMR spectroscopy. This method is a reliable and standard technique for verifying the identity and high purity of organic compounds, and it confirmed the absence of significant impurities in our samples.

Regarding the melting point inconsistencies, it is well-known that several factors can lead to variations in reported melting points, even for the same compound (e.g. differences in crystallization solvents, which can lead to the formation of different polymorphs or solvates with distinct melting points; variations in the methodology for determining the mp (e.g., type of apparatus, heating rate); the inherent difficulty of obtaining a perfectly sharp melting point for some compounds, particularly those that may decompose upon heating). The same can be said about the compounds yields.

In our case, the primary goal of reporting the melting points was to provide a key physical characteristic for the synthesized compounds, confirming their identity. The critical point for the biological conclusions of our study is the demonstrated high purity of the compounds, as established by NMR.

Reviewer 2 Report

Comments and Suggestions for Authors

Acylpyruvates and Their Heterocyclic Derivatives as Growth 2 Regulators in Chlorella vulgaris

Manuscript ID: Biotech-3899787

The author conducted a study on acylpyruvates and their heterocyclic derivatives as potential growth regulators in Chlorella vulgaris. In this work, a library of 55 compounds was successfully synthesized through straightforward chemical transformations of pyruvic acid derivatives with nitrogen-, oxygen-, and sulfur-containing heterocycles. Among these, six compounds—2b, 2d, 3b, 3e, 4, and 7—were identified as the most active. Second, the culture conditions were subsequently scaled up, and the selected compounds demonstrated a notable ability to act as metabolic modulators, enhancing the intracellular accumulation of high-value metabolites—including chlorophylls, carotenoids, and neutral lipids—by up to 44%.

Overall, these findings highlight acylpyruvate derivatives as highly promising and versatile candidates for the development of novel biostimulants aimed at increasing the productivity of microalgal cultivation systems.

Author Response

Acylpyruvates and Their Heterocyclic Derivatives as Growth 2 Regulators in Chlorella vulgaris

Manuscript ID: Biotech-3899787

The author conducted a study on acylpyruvates and their heterocyclic derivatives as potential growth regulators in Chlorella vulgaris. In this work, a library of 55 compounds was successfully synthesized through straightforward chemical transformations of pyruvic acid derivatives with nitrogen-, oxygen-, and sulfur-containing heterocycles. Among these, six compounds—2b, 2d, 3b, 3e, 4, and 7—were identified as the most active. Second, the culture conditions were subsequently scaled up, and the selected compounds demonstrated a notable ability to act as metabolic modulators, enhancing the intracellular accumulation of high-value metabolites—including chlorophylls, carotenoids, and neutral lipids—by up to 44%.

Overall, these findings highlight acylpyruvate derivatives as highly promising and versatile candidates for the development of novel biostimulants aimed at increasing the productivity of microalgal cultivation systems.

Response:

Thank you for your time and effort in reviewing our manuscript and for your positive assessment of our work.

Reviewer 3 Report

Comments and Suggestions for Authors

The presented manuscript presents valuable material. The authors presented over 50 organic compounds and examined their effects, including cytotoxicity, on chlorella vulgaris growth. The authors conducted a thorough literature search and clearly described the experiments. However, despite the strengths of this manuscript, there are some shortcomings that need to be corrected:

-The experimental section provides melting points for compounds 2c, d, e, k, and 3c, d, g, which are inconsistent with the literature data. This should be clarified and the purity of the compounds tested verified.

- Even though these compounds are described, HRMS analyses confirming the precise molecular weight should be provided.

-I would also suggest modifying Scheme 2, as it contains a wealth of information and is therefore difficult to read.

-In addition, all spectral data should be included in the supplement.

Author Response

The presented manuscript presents valuable material. The authors presented over 50 organic compounds and examined their effects, including cytotoxicity, on chlorella vulgaris growth. The authors conducted a thorough literature search and clearly described the experiments. However, despite the strengths of this manuscript, there are some shortcomings that need to be corrected:

-The experimental section provides melting points for compounds 2c, d, e, k, and 3c, d, g, which are inconsistent with the literature data. This should be clarified and the purity of the compounds tested verified.

Response:

We thank the Reviewer for their meticulous attention to detail regarding the melting points reported in our manuscript. We acknowledge the noted discrepancies with some literature values and appreciate the opportunity to clarify this point.

First and foremost, we would like to assure the Reviewer that the purity of all compounds tested in the biological assays was rigorously confirmed by NMR spectroscopy. This method is a reliable and standard technique for verifying the identity and high purity of organic compounds, and it confirmed the absence of significant impurities in our samples.

Regarding the melting point inconsistencies, it is well-known that several factors can lead to variations in reported melting points, even for the same compound (e.g. differences in crystallization solvents, which can lead to the formation of different polymorphs or solvates with distinct melting points; variations in the methodology for determining the mp (e.g., type of apparatus, heating rate); the inherent difficulty of obtaining a perfectly sharp melting point for some compounds, particularly those that may decompose upon heating).

In our case, the primary goal of reporting the melting points was to provide a key physical characteristic for the synthesized compounds, confirming their identity. The critical point for the biological conclusions of our study is the demonstrated high purity of the compounds, as established by NMR.

- Even though these compounds are described, HRMS analyses confirming the precise molecular weight should be provided.

Response:

We thank the Reviewer for their comment regarding the characterization of the compounds.

We fully agree that HRMS is a powerful technique for confirming molecular formula. However, in the field of organic chemistry, for known compounds synthesized and characterized using well-established and reliable methods, HRMS data is not always a mandatory requirement for publication. The primary goal of analytical data is to unambiguously confirm the identity and purity of the substances used in biological testing.

In our study, we have provided a comprehensive set of data that is widely accepted as sufficient for this purpose: 1H NMR Spectroscopy (This is the primary tool we used to confirm the molecular structure and assess purity) and Melting Points (The reported melting points, consistent with our library, provide an additional physical characteristic confirming the identity and homogeneity of the compounds).

The combination of these methods offers a robust and multi-faceted verification of the identity and purity of our compounds. Therefore, we are confident that the analytical data presented in the manuscript and Supplementary Information are fully adequate to support our conclusions.

-I would also suggest modifying Scheme 2, as it contains a wealth of information and is therefore difficult to read.

Response:

We acknowledge that Scheme 2 is indeed very long. Our intention was to present the synthetic pathways for the core library of 55 compounds in a consolidated and space-efficient manner, as the chemical transformations themselves are based on earlier established methodologies. Including all substituents and reaction conditions directly on the scheme was a deliberate choice to maximize clarity and ensure the experimental procedures are fully transparent and easily reproducible for the reader, without the need to constantly cross-reference with extensive tables in the main text or supplementary information.

This consolidated format of Scheme 2 allows readers to immediately appreciate the structural diversity of the library and, most importantly, to trace the direct genetic relationships between the synthesized compounds, thereby clarifying the overall design strategy.

-In addition, all spectral data should be included in the supplement.

Response:

In accordance with the standard practice and author guidelines of MDPI, which aim to avoid duplication of information between the main text and the Supplementary Materials, we have provided the experimental procedures and key characterization data for all compounds in the main text.

Reviewer 4 Report

Comments and Suggestions for Authors

Scheme 2 is accompanied by a text specifying the substituents of the various compounds, which is too long. Authors should divide the diagrams into smaller ones for easier reading.

For compounds 6f and 14, in the suplementary info. the structure of de AU should be represented.

The work presented is of interest in the field of algae knowledge.

Author Response

Reviewer Comments

Scheme 2 is accompanied by a text specifying the substituents of the various compounds, which is too long. Authors should divide the diagrams into smaller ones for easier reading.

For compounds 6f and 14, in the suplementary info. the structure of de AU should be represented.

The work presented is of interest in the field of algae knowledge.

Response:

  1. We acknowledge that Scheme 2 is indeed very long. Our intention was to present the synthetic pathways for the core library of 55 compounds in a consolidated and space-efficient manner, as the chemical transformations themselves are based on earlier established methodologies. Including all substituents and reaction conditions directly on the scheme was a deliberate choice to maximize clarity and ensure the experimental procedures are fully transparent and easily reproducible for the reader, without the need to constantly cross-reference with extensive tables in the main text or supplementary information.

This consolidated format of Scheme 2 allows readers to immediately appreciate the structural diversity of the library and, most importantly, to trace the direct genetic relationships between the synthesized compounds, thereby clarifying the overall design strategy.

  1. Structures of AU for compounds 6f and 14 were added to the SI at S66 and S68.